ecology, environmental science

pollinators, temporal stability, crops, insect diversity, inter-annual variation, dominant species

**Author for correspondence:**
Deepa Senapathi
e-mail: g.d.senapathi@reading.ac.uk

# Wild insect diversity increases inter-annual stability in global crop pollinator communities

Deepa Senapathi[1], Jochen Fründ[2], Matthias Albrecht[3], Michael P. D. Garratt[1], David Kleijn[4], Brian J. Pickles[5], Simon G. Potts[1], Jiandong An[6], Georg K. S. Andersson[7], Svenja Bänsch[8,9], Parthiba Basu[10], Faye Benjamin[11], Antonio Diego M. Bezerra[12], Ritam Bhattacharya[10], Jacobus C. Biesmeijer[13], Brett Blaauw[14], Eleanor J. Blitzer[15], Claire A. Brittain[16], Luísa G. Carvalheiro[17,18], Daniel P. Cariveau[19], Pushan Chakraborty[10], Arnob Chatterjee[10], Soumik Chatterjee[10], Sarah Cusser[20], Bryan N. Danforth[21], Erika Degani[1], Breno M. Freitas[12], Lucas A. Garibaldi[7,22], Benoit Geslin[23], G. Arjen de Groot[24], Tina Harrison[25], Brad Howlett[26], Rufus Isaacs[27,28], Shalene Jha[29], Björn Kristian Klatt[9,30], Kristin Krewenka[31], Samuel Leigh[1], Sandra A. M. Lindström[30,32,33], Yael Mandelik[34], Megan McKerchar[35], Mia Park[21,36], Gideon Pisanty[37], Romina Rader[38], Menno Reemer[13], Maj Rundlöf[30], Barbara Smith[10,39], Henrik G. Smith[40], Patrícia Nunes Silva[41], Ingolf Steffan-Dewenter[42], Teja Tscharntke[9], Sean Webber[1], Duncan B. Westbury[35], Catrin Westphal[8,9], Jennifer B. Wickens[1], Victoria J. Wickens[1], Rachael Winfree[11], Hong Zhang[6] and Alexandra-Maria Klein[43]

[1]Centre for Agri-Environmental Research, School of Agriculture, Policy & Development, University of Reading, Reading, UK
[2]Biometry and Environmental System Analysis, Faculty of Environment and Natural Resources, University of Freiburg, Freiburg, Germany
[3]Institute for Sustainability Sciences, Agroscope, Zurich, Switzerland
[4]Plant Ecology and Nature Conservation Group, Wageningen University, Wageningen, The Netherlands
[5]School of Biological Sciences, University of Reading, Reading, UK
[6]Institute of Apicultural Research, Chinese Academy of Agricultural Sciences, Beijing, People's Republic of China
[7]Universidad Nacional de Río Negro, Instituto de Investigaciones en Recursos Naturales, Agroecología y Desarrollo Rural, Río Negro, Argentina
[8]Functional Agrobiodiversity, Department of Crop Sciences, University of Göttingen, Göttingen, Germany
[9]Agroecology, Department of Crop Sciences, University of Göttingen, Göttingen, Germany
[10]Centre for Pollination Studies, University of Calcutta, Kolkata, India
[11]Department of Ecology, Evolution and Natural Resources, Rutgers, The State University of New Jersey, New Brunswick, USA
[12]Setor de Abelhas, Departamento de Zootecnia, Universidade Federal do Ceará, Fortaleza - CE, Brazil
[13]Naturalis Biodiversity Centre, Leiden, The Netherlands
[14]Department of Entomology, University of Georgia, Athens, Georgia, USA
[15]Department of Biology, Carroll College, Harrison Helena, USA
[16]Syngenta, Jealott's Hill International Research Centre, Bracknell, Berkshire RG42 6EY, UK
[17]Departamento de Ecologia, Universidade Federal de Goiás, Campus Samambaia, Goiânia, Brazil
[18]Centre for Ecology, Evolution and Environmental Changes (cE3c), University of Lisboa, Lisbon, Portugal
[19]Department of Entomology, University of Minnesota, St Paul, USA
[20]W. K. Kellogg Biological Station, Michigan State University, MI, USA
[21]Department of Entomology, Cornell University, Ithaca, NY, USA
[22]Consejo Nacional de Investigaciones Científicas y Técnicas. Instituto de Investigaciones en Recursos Naturales, Agroecología y Desarrollo Rural, San Carlos de Bariloche, Río Negro, Argentina

[23]IMBE, Aix Marseille Univ, Avignon Université, CNRS, IRD, Marseille, France

[24]Wageningen Environmental Research, Wageningen University and Research, Wageningen, The Netherlands

[25]Department of Entomology and Nematology, University of California Davis, Davis, USA

[26]The New Zealand Institute for Plant & Food Research Limited, New Zealand

[27]Department of Entomology, Michigan State University, East Lansing, USA

[28]Ecology, Evolutionary Biology, and Behavior Program, East Lansing, USA

[29]Department of Integrative Biology, The University of Texas at Austin, USA

[30]Department of Biology, Biodiversity, Lund University, Lund, Sweden

[31]Heidelberg Research Service, University of Heidelberg, Heidelberg, Germany

[32]Department of Ecology, Swedish University of Agricultural Sciences, Uppsala, Sweden

[33]Swedish Rural Economy and Agricultural Society, Kristianstad, Sweden

[34]Department of Entomology, The Robert H. Smith Faculty of Agriculture, Food and Environment, The Hebrew University of Jerusalem, Rehovot, Israel

[35]School of Science and Environment, University of Worcester, Worcester, UK

[36]Field Engine Wildlife Research and Management, Moodus, CT 06469, USA

[37]Agriculture and Agri-Food Canada, Canadian National Collection of Insects, Arachnids and Nematodes, Ontario, Canada

[38]School of Environment and Rural Science, University of New England, Armidale, Australia

[39]Centre for Agroecology, Water and Resilience, Coventry University, UK

[40]Centre of Environmental and Climate Research & Department of Biology, Lund University, Sweden

[41]Programa de Pós-Graduação em Biologia, Universidade do Vale do Rio dos Sinos (UNISINOS), Av. Unisinos, 950, São Leopoldo, RS, Caixa Postal 93022-750, Brazil

[42]Department of Animal Ecology and Tropical Biology, University of Würzburg, Würzburg, Germany

[43]Nature Conservation and Landscape Ecology, Faculty of Environment and Natural Resources, University of Freiburg, Freiburg, Germany

DS, 0000-0002-8883-1583; JF, 0000-0002-7079-3478; MA, 0000-0001-5518-3455; MPDG, 0000-0002-0196-6013; DK, 0000-0003-2500-7164; BJP, 0000-0002-9809-6455; SGP, 0000-0002-2045-980X; JA, 0000-0002-4203-4554; GKSA, 0000-0002-9669-6895; SB, 0000-0001-7332-7213; PB, 0000-0002-8942-0161; ADMB, 0000-0002-8070-5582; JCB, 0000-0003-0328-0573; BB, 0000-0001-6165-2713; EJB, 0000-0002-6606-1741; CAB, 0000-0003-1104-2946; LGC, 0000-0001-7655-979X; DPC, 0000-0002-3064-0071; SC, 0000-0002-0100-026X; BND, 0000-0002-6495-428X; ED, 0000-0002-8000-8744; BMF, 0000-0002-9932-2207; LAG, 0000-0003-0725-4049; BG, 0000-0002-2464-7998; GAdG, 0000-0001-7308-9200; TH, 0000-0001-8735-0289; BH, 0000-0002-0694-135X; RI, 0000-0001-7523-4643; SJ, 0000-0001-7199-6106; BKK, 0000-0001-8241-6445; KK, 0000-0002-1913-1057; SL, 0000-0003-4102-0537; SAML, 0000-0002-8403-3509; YM, 0000-0002-9576-119X; MP, 0000-0002-1250-7664; GP, 0000-0003-2076-430X; RR, 0000-0001-9056-9118; MR, 0000-0003-1732-4047; MR, 0000-0003-3014-1544; BS, 0000-0002-0506-0331; HGS, 0000-0002-2289-889X; PNS, 0000-0002-9215-9822; IS-D, 0000-0003-1359-3944; TT, 0000-0002-4482-3178; DBW, 0000-0001-7094-0362; CW, 0000-0002-2615-1339; JBW, 0000-0003-0475-6780; VJW, 0000-0002-2295-0635; RW, 0000-0002-1271-2676; HZ, 0000-0003-0939-9348; A-MK, 0000-0003-2139-8575

While an increasing number of studies indicate that the range, diversity and abundance of many wild pollinators has declined, the global area of pollinator-dependent crops has significantly increased over the last few decades. Crop pollination studies to date have mainly focused on either identifying different guilds pollinating various crops, or on factors driving spatial changes and turnover observed in these communities. The mechanisms driving temporal stability for ecosystem functioning and services, however, remain poorly understood. Our study quantifies temporal variability observed in crop pollinators in 21 different crops across multiple years at a global scale. Using data from 43 studies from six continents, we show that (i) higher pollinator diversity confers greater inter-annual stability in pollinator communities, (ii) temporal variation observed in pollinator abundance is primarily driven by the three-most dominant species, and (iii) crops in tropical regions demonstrate higher inter-annual variability in pollinator species richness than crops in temperate regions. We highlight the importance of recognizing wild pollinator diversity in agricultural landscapes to stabilize pollinator persistence across years to protect both biodiversity and crop pollination services. Short-term agricultural management practices aimed at dominant species for stabilizing pollination services need to be considered alongside longer term conservation goals focussed on maintaining and facilitating biodiversity to confer ecological stability.

## 1. Introduction

The crucial role played by pollinators in the reproduction of flowering plants is well-established [1]. Biotic pollination is important for the reproduction of at least 78% of wild plants [2] and insects contribute to the pollination of 75% of leading global crops [3]. Crop systems have also recently become more pollinator dependent because of a disproportionate increase in the area cultivated with entomophilous flowering crops [4]. Given the documented declines of wild insect pollinators in some NW European and North American landscapes where these crops are grown [1,5,6], understanding temporal variation in assemblages is important to maintain ongoing food security.

Higher pollinator diversity can lead to increases in fruit and seed set in plants and is an important predictor of crop yields worldwide [7,8]. Conversely, pollinator communities with lower diversity and fewer species have been linked to lower fruit set or seed production, and decreased temporal and spatial stability within seasons [9–11], and may be one reason for lower inter-annual stability of yields in pollinator-dependent crops [1]. While biologically diverse communities can enhance ecological resilience [12,13], and diversity is a key factor affecting system stability [14], most ecological communities are generally made up of a few species that are numerically abundant and may have many rarer species with very few individuals [15].

Evidence suggests that numerically dominant species may provide most of the ecosystem services [16], with Kleijn *et al.* [17] finding that approximately 80% of biotic crop pollination in Europe and North America are fulfilled by approximately 2% of the pollinator species in a community. In addition, the scale of spatial assessment, is also important, with Winfree *et al.* [18] showing that the number of wild bee species required for reaching a minimum pollination service threshold rapidly increased with spatial scale, indicating that maintaining pollination services across large areas requires many species, including rare ones. Providing stable pollination services for crop systems across several years is needed for sustainable crop production, but the mechanisms driving temporal stability for ecosystem functioning and services still remains an important but poorly understood phenomenon [19].

Previous studies aimed at disentangling the mechanisms of temporal stability highlight the role of both diversity and

dominance. Lehman & Tilman [20] showed that greater diversity increases the temporal stability of the entire community but decreases the temporal stability of individual populations. The counterview is that dominant species, rather than diversity itself, might regulate temporal stability—for e.g. Sasaki & Lauenroth [21] found that temporal stability in a shortgrass steppe plant community was controlled by dominant species rather than by community diversity. In addition, species asynchrony has also been considered an important mechanism of diversity–stability relationships and may lead to higher stability on the community level even when the stability of individual populations decreases with diversity. However, the majority of such studies have used long-term observations of the same plant communities over time (for e.g. [22]), while such equivalent information on pollinators in general or even crop pollinator communities in particular are lacking.

A few multi-year, single-crop studies exist showing that pollinator communities can vary over longer time periods [9,23,24]. What implications this may have for stability remains unknown due to a lack of synthesized knowledge on temporal dynamics of crop pollinator communities and underlying driving factors. For example, evidence for the contribution of managed pollinators to the temporal stability of the overall crop pollinator community is largely lacking. Such knowledge gaps, if addressed, could lead to a better understanding of the stability and long-term resilience of global crop systems that rely on insect pollination. Temporal stability of ecosystem functioning increases predictability and reliability of ecosystem services and understanding the drivers of stability across spatial scales is important for land management and policy decisions [25].

Here, we synthesize data from multiple studies to examine factors that affect the temporal stability of crop pollinator communities, which in turn has implications for the stability of pollination services provided. Using data from 43 studies across six continents, we characterize the annual variation observed in crop pollinators and explore the following questions: (1) is temporal stability of crop pollinator communities primarily driven by the diversity of pollinator communities or by inter-annual stability of dominant species? (2) What crop characteristics if any (e.g. annual/perennial, flower morphology, mass flowering/non-mass flowering crops) influence inter-annual stability of crop pollinator communities? (3) Does inter-annual variation observed in pollinator communities differ between climatic regions (i.e. tropics and temperate study areas)?

## 2. Material and methods

### (a) Data collection

We collated datasets from 12 countries across six continents on 21 crop species to examine the variations in richness and abundance of insect pollinators in crop systems. The criteria for inclusion in the analyses were as follows: data on crop pollinator species/morpho-species were required (a) from the same crop for two or more years, (b) with consistent sampling methods used across years, (c) focused on flower visitation data and (d) in the case of annual crops, field sites were required to be within 500 m of the crop field used for recording in previous years to make sure they could be visited by the same pollinator communities. Our final dataset included information on 375 crop fields (hereafter referred to as sites) from 43 studies (see electronic supplementary material, table S1 for additional information).

Data were standardized to ensure that species names and taxonomic groups were consistent across all studies prior to analyses.

Each dataset was classified on the basis of climatic region (tropical/temperate), crop type (annual/perennial), plant family and flower type (open/not open)—based on nectar accessibility criteria in Garibaldi *et al.* [26]. In addition, we distinguished crop species that exhibit mass flowering (MFC)—i.e. short duration intense bloom with high floral density, from those with extended flowering periods with lower density and more sparse blooms. Some crops are clearly defined as mass flowering in the literature [27–31], while others remain ambiguous. To overcome this uncertainty, we requested the original authors to indicate if their crop was considered as MFC in their study and that is reflected in the dataset and subsequent analyses (see electronic supplementary material, table S2). Almonds (*Prunus dulcis*), apples (*Malus domestica*), highbush blueberry (*Vaccinium corymbosum*), cranberry (*Vaccinium angustifolium*), red clover (*Trifolium pratense*), field beans (*Vicia faba*), oilseed rape or canola (*Brassica napus*), pears (*Pyrus communis*), pak choi (*Brassica rapa* subsp. *chinensis*) and turnips (*Brassica rapa* subsp. *rapa*) were classified as MFC. Non-mass flowering crops in our analyses include avocado (*Persea americana*), bitter gourd (*Momordica charantia*) and brinjal (*Solanum melongena*)—also known as eggplant or aubergine, cashew (*Anacardium occidentale*), cotton (*Gossypium hirsutum*), kiwifruit (*Actinidia deliciosa*), mango (*Mangifera indica*), mustard (*Brassica napus*), onion (*Allium cepa*), pumpkin (*Cucurbita pepo*), ridge gourd (*Luffa acutangula*), spine gourd (*Momordica dioica*), strawberry (*Fragaria x ananassa*) and watermelon (*Citrullus lanatus*). Note: *Brassica napus* includes oilseed rape (OSR)—a MFC in Europe and North America but a different subspecies considered as a type of mustard in India which is not grown as MFC.

### (b) Characterizing year to year variation in crop pollinators

Initially, crop pollinators recorded were classified into taxonomic groups which included the following: (i) honeybees (including *Apis mellifera*, *Apis cerana*, *Apis dorsata* and other recorded as *Apis* sp.); (ii) bumblebees (all *Bombus* sp.); (iii) other bees (wild solitary and social bees including stingless bees but excluding bumblebees and honeybees); (iv) butterflies and moths; (v) hoverflies; (vi) other Diptera (flies excluding hoverflies); (vii) wasps; and (viii) beetles. The single most dominant taxonomic group and species were identified at all study sites (figure 1) based on recorded abundance and a binary (change/no change) analysis was used to determine whether the most dominant group and species remained constant across all years of sampling.

To characterize between year variation in crop pollinators, (i) a coefficient of variation (CV) of total pollinator abundance and (ii) a CV of pollinator species richness was calculated for each site across all years of the study. The CV (which incorporates a bias correction) is defined as the ratio of the sample standard deviation 's' to the sample mean x̄—i.e. $CV = s/\bar{x}$ and shows the extent of variability in relation to the mean of the population. These measures were calculated using species level data for each study site and the mean and standard deviation of these two measures were also calculated for each individual study (electronic supplementary material, figure S1). In addition, the CV of abundance and CV of richness were calculated for each site for every pairwise year comparisons (i.e. Y1&Y2; Y2&Y3; Y3&Y4 etc.) to account for studies having a different number of years of data.

Other calculated indices included (a) CV of honeybee abundance, (b) CV of the proportion of honeybees, (c) CV of the most dominant pollinator species across all years and (d) the mean Shannon index of pollinator diversity (H′) were calculated across years. The Shannon diversity index was chosen as it accounts for the evenness of the species present, thus reflecting effective diversity, and is less sensitive to sampling effects than

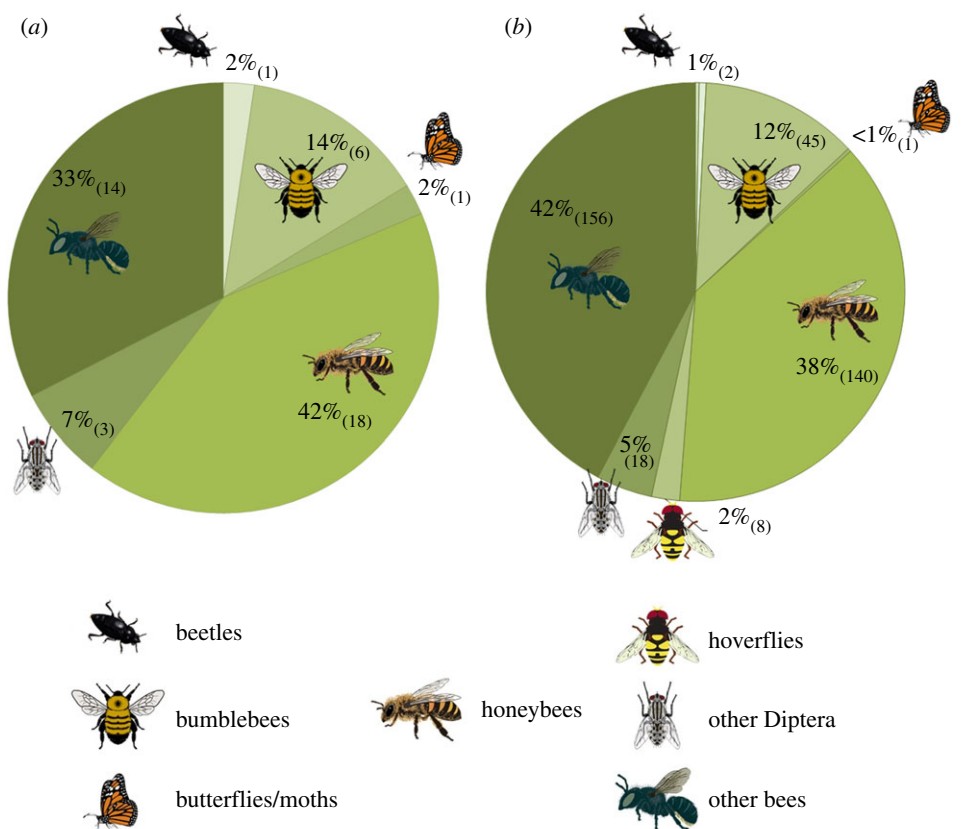

**Figure 1.** Most dominant taxonomic group of crop pollinators across years at (*a*) study and (*b*) site levels with number of studies and number of sites in parentheses.

species richness [32,33]. Since a subset of studies (28 out of 43) also recorded temperature at the study sites, a standard deviation (s.d.) of temperature was also calculated as a measure of variation in local climatic condition across years.

## (c) Factors influencing the observed variation

In order to examine the potential drivers of inter-annual variation in crop pollinator communities, linear mixed-effects models were constructed using (i) CV of total pollinator abundance and (ii) CV of pollinator species richness. These two indices were calculated across all years of each study and for every pairwise year in each study to account for studies with different numbers of years of observations and ensure checks for sensitivity and robustness. The models included descriptors of pollinator communities such as Shannon diversity (H′) of pollinators, CV of dominant species, and change in dominant pollinator species between years (Y/N) as fixed effects. External predictors including climatic region (tropical/temperate), crop type (annual/perennial), crop family, flower type, MFC (Y/N) and SD of site temperature were also included as other explanatory variables. Study ID was included in all models as a random effect and for models where the response variables were calculated for every two years of the study, site ID nested within the study ID were used as random effects (and identified as relevant indicated by positive variance estimates).

The calculated indices were tested for collinearity and correlated variables were not used within the same models (see correlation matrix in electronic supplementary material, table S3). Similarly, categorical predictors which exhibited significant collinearity were not used as variables within the same models. A series of candidate models were constructed for each response variable. Each candidate model was 'dredged' to obtain a series of plausible intermediate models. Intermediate models with Δ AICc value less than 7 of the model with lowest AICc were averaged (using the default zero average method) to obtain the final outputs. Residual plots for final models were used to check for heteroscedasticity. Models were fitted using maximum likelihood (ML) and

analysed using the 'lme4' [34] and 'MuMIn' [35] packages. All statistical analyses were carried out in R v. 4.0.3 statistical software [36].

## (d) Influence of dominant pollinator species

To test whether variation in total crop pollinator abundance was driven primarily by variation of the most dominant pollinator species, a paired t-test was used to determine whether the CV of total pollinator abundance was significantly different from the CV of the abundance of the single most dominant pollinator species. The same test was repeated using the combined CV of the abundance of the two-most, three-most and four-most dominant species to determine how many dominant pollinator species were required to influence the overall variation in total abundance observed. While abundance of dominant species will always be a subset of the total pollinator abundance, these tests were conducted to determine how many dominant species it took to match the change in overall pollinator abundance across years and determine the minimum number of species that drive the temporal variation in overall pollinator abundance. A Welch two-sample t-test was used to determine if inter-annual variation in pollinator abundance differed between sites dominated by honeybees versus other pollinator species. Sites where there was mixed dominance between honeybees and other pollinators were excluded from this analysis.

## (e) Species dominance and stability effect

To further understand mechanisms of stability and particularly the relationship of the dominant species to the whole community, we calculated the correlation between the changes in abundance of the dominant species and the changes in abundance of the rest of the community. Negative correlation (negative covariance) suggests asynchrony, which is considered a key driver of stability and the main mechanism of diversity–stability relationships [37]. Negative correlations could indicate density compensation or different responses to environmental variation [12]. In general, the higher the asynchrony (i.e. more negative the correlation),

**Table 1.** The proportion of studies and sites showing inter-annual changes in the dominant taxonomic groups and species of crop pollinators; actual no. of studies and sites shown within parentheses. Note: one study with five sites (Pisa01) had only morpho-species level data.

|  | study level | | site level | |
|---|---|---|---|---|
|  | change | no change | change | no change |
| taxonomic group | 27.9% (12) | 72.1% (31) | 31.2% (117) | 68.8% (258) |
| species (excl Pisa01) | 48.1% (20) | 51.2% (22) | 50.8% (188) | 49.2% (182) |

the stronger the contribution to stability. With our short time series, many correlations are −1 or +1, without an even continuous gradient in the degree of asynchrony. Therefore, we separated sites by asynchronous ($r \leq 0$) or synchronous ($r > 0$) fluctuations of the dominant pollinator species in comparison to the rest of the pollinator community and, for each group separately, repeated the paired t-test of the CV of the dominant species versus the whole community.

## 3. Results

### (a) Characterizing year to year variation in crop pollinators

Honeybees were dominant across 41.9% of studies with other wild bees (32.6%) representing the next most dominant group (figure 1*a*). At the site level, other wild bees were the most dominant group at 41.6%, with honeybees (38.0%) the second most dominant (figure 1*b*). The dominant taxonomic group did not change between years in most of the studies or the sites, whereas the dominant species varied between years in approximately half the studies and half the sites (table 1). The mean (±s.d.) of the CV of total pollinator abundance, and the CV of total pollinator richness for all sites within each study are provided in electronic supplementary material, figure S1.

### (b) Factors influencing the observed variation

The relative variability of total pollinator abundance across all years was significantly related to the Shannon diversity (table 2, estimate = −0.16, $z = 3.96$, $p < 0.0001$, figure 2*a*). It was also significant whether the most dominant species varied between years: systems where dominant species stayed the same showed less inter-annual variation in overall pollinator abundance (table 2, estimate = −0.08, $z = 2.23$, $p = 0.03$, figure 2*b*). However, in models using CV of abundance for every two years, the variability in dominant species showed no significant relationship (table 2, estimate = −0.05, $z = 1.42$, $p = 0.16$)

Having a diverse pollinator community also reduced the inter-annual variation in pollinator species richness (table 2, estimate = −0.16, $z = 5.61$, $p < 0.0001$, figure 3*a*), and this was true for indices calculated across all years of the studies as well as every two years of the studies (table 2). The relative change in species richness between years was related to the

change in the abundance of the most dominant species, with study systems showing larger changes in species richness if there was increased inter-annual variation in dominant species abundance across all years (table 2, estimate = 0.09, $z = 3.31$, $p < 0.001$, figure 3*b*). This was also significant in models accounting for change in species richness for every two years (table 2, estimate = 0.12, $z = 3.77$, $p < 0.001$). However, any change in dominant species across years showed no significant relationship with the relative change in species richness. The change in pollinator species richness also varied between climatic regions with crops grown in temperate systems showing less inter-annual variability in pollinator species richness than crops in tropical areas (table 2, figure 3*c*).

Other factors tested including crop family, flower type, annual versus perennial crop type, mass flowering or site temperature did not show any significant relationship with variability observed in the abundance or richness of species across years.

### (c) Influence of dominant pollinator species

It took the pooled abundance of the three-most dominant pollinator species to match the relative variability of total pollinator abundance (respective mean CVs: 0.58 versus 0.55, $t = 1.09$, d.f. = 362, $p = 0.2$, difference in means = 0.03). The relative variability of total pollinator abundance at the site level was found to be significantly lower than that of the single ($t = 9.56$, d.f. = 362, $p$-value < 0.001, difference in means = 0.17) and top two-most dominant species ($t = 6.34$, d.f. = 362, $p$-value < 0.001, difference in means = 0.07). Sites where honeybees were dominant species (mean CV = 0.46) were found to have significantly lower variability ($t = 3.25$, d.f. = 295.26, $p = 0.001$) than sites where other bees were dominant species (mean CV = 0.60).

Where the dominant species changed asynchronously to the rest of the community, the difference between the CV of the dominant species and CV of total abundance was strong, with less than half the variability in the whole community than in the dominant species ($t = −11.02$, d.f. = 125, $p$-value less than 0.0001, mean of total 0.31, mean of single most dominant species 0.67; difference in means = 0.36). By contrast, CV of total abundance was only slightly lower than the CV of the dominant species where the dominant species changed synchronously with the rest of the community ($t = −3.48$, d.f. = 219, $p$-value = <0.001, mean of total 0.65, mean of single most dominant species 0.71, difference in means = 0.06, figure 4). In simple terms, the stability of the whole pollinator community only increased to a considerable degree when other species buffered changes by asynchronous fluctuations.

## 4. Discussion

This study is the first to use a global dataset to explore inter-annual variation in crop pollinator communities and has revealed several important features of community stability. Our findings indicate that: (i) crop pollinator communities with higher pollinator diversity are more stable between years, and (ii) the variation observed in pollinator communities is driven by dominant species changes across years. The importance of other species in addition to the stability of the dominant species was in line with mechanisms of

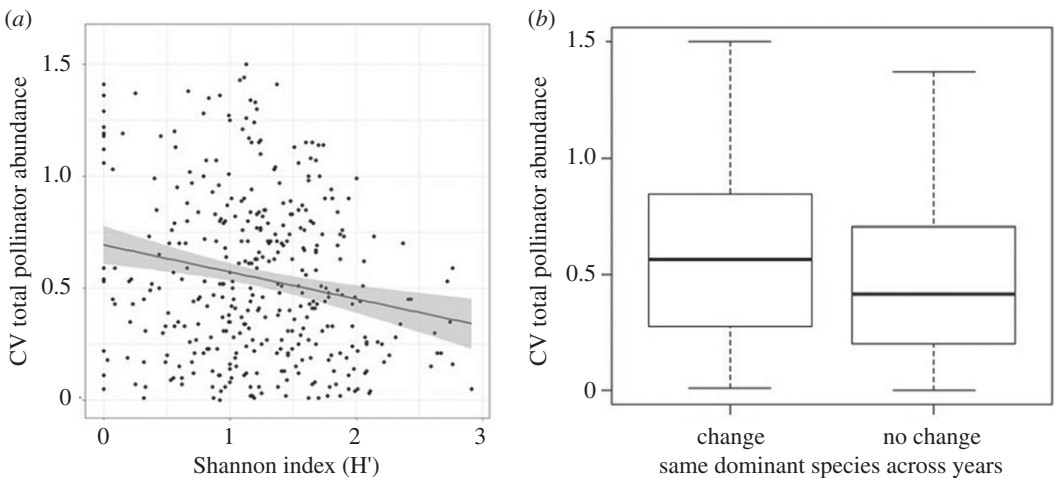

**Figure 2.** The relative change in total abundance of crop pollinators between years are driven by (*a*) species diversity (Shannon index) with 95% CI and (*b*) the change in dominant species.

**Table 2.** Results of model averaging of candidate models that were within AICc Δ7 of the model with the lowest AICc value.

| response variable | fixed effects remaining in the averaged model | estimate | adjusted SE | z-value | p-value |
|---|---|---|---|---|---|
| CV total pollinator abundance | **models with CV calculated across all years of the studies** | | | | |
| | conditional R² = 0.33; marginal R² = 0.09 | | | | |
| | same dominant species | −0.08482 | 0.03802 | 2.231 | 0.0257* |
| | H′ index | −0.15584 | 0.03932 | 3.964 | $7.38 \times 10^{-5}$*** |
| | climatic region | 0.08302 | 0.09064 | 0.916 | 0.3598 |
| | MFC | −0.08627 | 0.08326 | 1.036 | 0.3001 |
| | **models with CV calculated for every two years of the studies** | | | | |
| | conditional R² = 0.35; marginal R² = 0.06 | | | | |
| | same dominant species | −0.05286 | 0.03726 | 1.418 | 0.15607 |
| | H′ index | −0.10368 | 0.03792 | 2.734 | 0.00626** |
| | climatic region | 0.11703 | 0.08691 | 1.347 | 0.17812 |
| | MFC | −0.10889 | 0.03726 | 1.322 | 0.18609 |
| CV of pollinator species richness | **models with CV calculated across all years of the studies** | | | | |
| | conditional R² = 0.56; marginal R² = 0.19 | | | | |
| | climatic region | 0.16877 | 0.08576 | 1.968 | 0.049096* |
| | CV of most dominant species[a] | 0.09774 | 0.02957 | 3.305 | 0.000951*** |
| | H′ index | −0.16173 | 0.02879 | 5.616 | $<2 \times 10^{-16}$*** |
| | MFC | 0.00435 | 0.11645 | 0.037 | 0.970190 |
| | **models with CV calculated for every two years of the studies** | | | | |
| | conditional R² = 0.37; marginal R² = 0.09 | | | | |
| | climatic region | 0.111412 | 0.079390 | 2.138 | 0.032545* |
| | CV of most dominant species[a] | 0.121180 | 0.032136 | 3.771 | 0.000163*** |
| | H′ index | −0.048424 | 0.037559 | 2.242 | 0.024961* |
| | MFC | 0.002177 | 0.051874 | 0.073 | 0.942094 |

[a]CV of most dominant species remained significant when it was the single most dominant, two-most dominant as well as three-most dominant species.

diversity–stability relationships: while the stability of the dominant species was similar to the total community where the dominant species fluctuated synchronously with the rest of the community, community abundance was much more stable than abundance of the dominant species where these fluctuations were asynchronous. Neither the variation

in abundance nor the variation in species richness was significantly affected by any crop characteristics.

Our results show that sites with higher pollinator species diversity experience less variation in total crop pollinator abundance and less change in pollinator species richness between years. These results concur with studies from

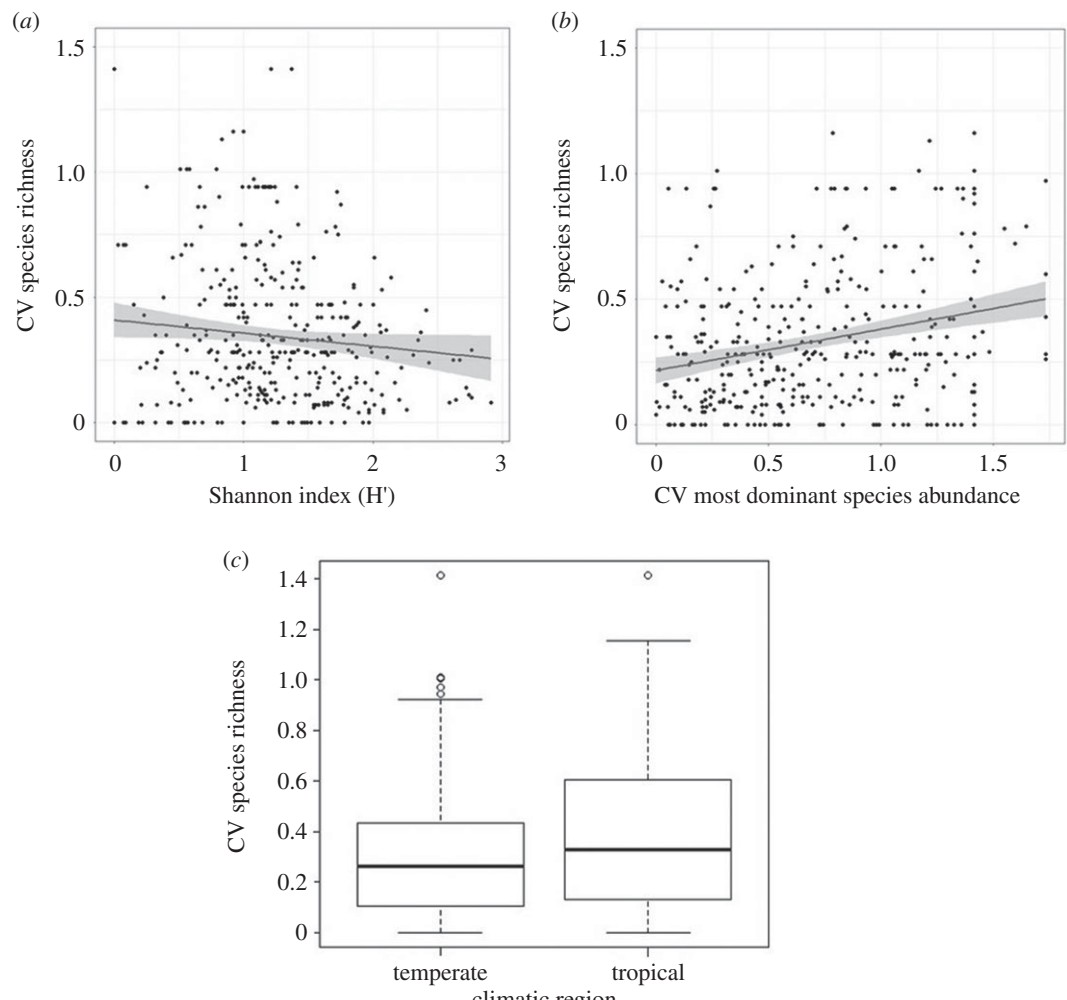

**Figure 3.** Inter-annual variability of crop pollinator species richness is driven by (*a*) relative change in the abundance of the most dominant species (showing 95% CI), (*b*) average species diversity (Shannon index) and (*c*) climatic region.

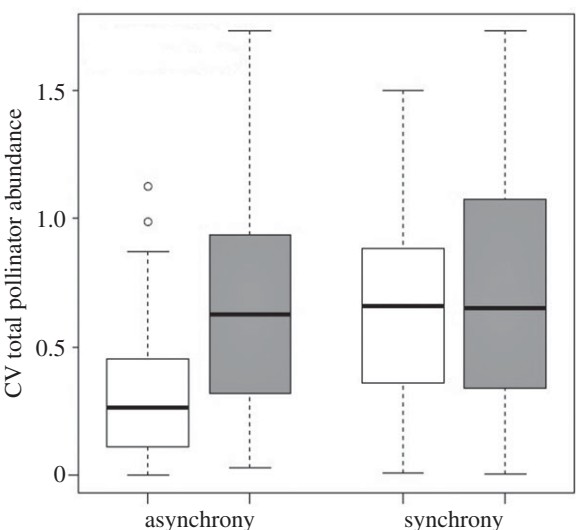

**Figure 4.** Relative change in single most dominant species (grey) compared to relative change in overall abundance (white) when split into asynchronous (left side) and synchronous (right side) pollinator communities.

individual cropping systems which have shown that diversity provides greater spatial and temporal stability and resilience [12,23], and support the theory that ecological systems with higher species diversity are better buffered against inter-annual variation in species abundance, and possibly more

resilient to changes in the longer term [14]. This has implications beyond ecological resilience, as stable pollination services could help mitigate risks and uncertainties for farmers growing pollinator-dependent crops, providing economic resilience.

In addition to diversity, our results demonstrate that dominant species play a significant role in the inter-annual stability of crop pollinator communities. Honeybees were found to be the single most dominant species in 18 out of 43 datasets and in 140 out of 375 sites which concurs with the findings of Kleijn *et al.* [17]. Sites where honeybees were the dominant species across all years also showed greater inter-annual stability in abundance when compared to sites dominated by other species. Unlike wild pollinators, managed pollinators are often placed near crops and due to hive management practices may show less variability in abundance between years. Managed pollinators are considered to supplement rather than substitute pollination by wild insects in most crop pollination systems [38], but there is experimental evidence to suggest that managed bees in high numbers could displace wild pollinators from crop fields [39]. Our study systems from Argentina, for instance, were entirely reliant on managed *Apis mellifera* and no other species were recorded. The management of bees could therefore be an important contributor to the inter-annual variability observed in the crop pollinator community depending on the placement of hives, stocking densities

and how much these vary from one year to the next. Careful targeting of managed pollinators could be used to increase the stability of pollination [40–42], particularly in those crops for which inter-annual variation is high due to fluctuations in populations of the dominant wild pollinators.

While we can say with a high level of certainty that most honeybees recorded in the USA and European studies were from managed hives, it is difficult to distinguish between managed and wild honeybees in other studies. For example, in China and India, while almost all *Apis mellifera* were managed and all *Apis dorsata* wild, it is difficult to distinguish between wild and managed *Apis cerana* with any degree of certainty. In addition, certain areas—particularly in Western Europe—use *Bombus terrestris* as a managed pollinator, and managed and wild individuals of this species are indistinguishable from each other. Therefore, we cannot draw specific conclusions on the effect of managed pollinators on the changes in richness and turnover of wild pollinator communities but raise this as a possible question to be explored in future studies.

From our results, we also infer that a significant part of the year to year variation in crop pollinator abundance is driven by as few as three of the most dominant species within each system (see list of dominant species by study in electronic supplementary material, table S4). This is consistent with the findings of Kleijn *et al.* [17] who showed that the three-most dominant pollinator species account for two-thirds of flower visits recorded. Even if only a few species are quantitatively important in crop pollination systems, enhancing stability by managing for diversity effects delivered through asynchrony among species could be really effective as our results above have indicated. It is worth noting that while the delivery of crop pollination services may be predominantly driven by a few key functional pollinator species [17], depending on the context, the diversity and abundance of other pollinators may complement or largely replace the functional role of dominant species [43].

The Winfree *et al.* [18] study—which explored functional consequences of spatial turnover in crop pollinator communities—indicated that more species would be required to fulfil the minimum pollination service threshold if dominance effects were to be removed or lost, but that is based on the assumption that another species would be unable to take over the dominant role through increased abundance. This raises questions of which systems would remain resilient in the event these specific dominant species are lost due to future environmental conditions. For example, field beans flower morphology excludes small solitary bees and depends predominantly on effective flower visits from long-tongued bumblebees [44,45], may be less resilient to the loss of dominant pollinators when compared to crops like oilseed rape dependent on a diverse suite of pollinators [44].

While no effect of the climatic region was observed on the inter-annual change in pollinator abundance, there was less variation in pollinator species richness in temperate crops than in crops grown in the tropics. Studies from temperate regions ($n = 29$) showed a higher average Shannon diversity (H′ = 1.21) than studies from the tropics ($n = 13$, H′ = 1.19) but the difference was not statistically significant (electronic supplementary material, figure S2, $t = 0.26$, d.f. = 356, $p = 0.74$), and it is difficult to disentangle whether this result may be due to differences in sampling effort. The difference between the temperate and tropical studies could not be attributed to contrasting temperature regimes in the different

climatic regions as we did not detect a significant effect of temperature on inter-annual stability of crop pollinators in any of the models. Pollinator populations are known to be sensitive to weather conditions [31] with temperature influencing pollinator phenology [46] as well as plant–pollinator interactions [47]. Our analyses indicated that the crops in the tropics experienced significantly less variation in temperature than those in temperate regions ($t = 6.71$; d.f. = 34.74; $p < 0.001$, electronic supplementary material, figure S3) but insufficient climate data across all the datasets (only 28 studies of the 43 recorded temperature), meant this aspect could not be fully explored within this study.

Of the 43 studies used, 25 studies had two years of data, 14 studies three years of data and four studies with four or more years of repeated sampling. With these differences in number of years of sampling, our global synthesis has only provided a first step to looking at temporal dynamics. Estimates of temporal dynamics may vary with the number of years sampled and every effort has been made to account for these differences by analysing changes in observed in every two years of each study. It is to be noted that results of the models with the pairwise year calculations were consistent with the model using data across all the years, but further measures to account for any differences caused by a varying number of sampling years, and are beyond the scope of this manuscript. Also, the diversity–stability effect identified, may be linked to sampling effort with lower sampling leading to high CV values and low diversity between years. As this is a collated dataset consisting of various studies that have taken place across several geographic regions across multiple years and we cannot retrospectively change the sampling effort, we acknowledge that the CV may be sensitive to these underlying effects and raise this as a point to be considered in future studies.

Many studies to date have focused on spatial variations observed between crops, fields and across different landscapes [29,48,49], while relatively few studies have considered temporal variation caused by differences in crop flowering times [31,40,50] and even these focussed only on within-season variation. To the best of our knowledge, our study is the first to explore temporal variation in pollinator communities across different crops. Our results highlight the importance of considering both wider pollinator diversity as well as the abundance of dominant species in understanding the inter-annual stability of crop pollinators. Temporal stability of ecosystem functioning increases the predictability and reliability of ecosystem services and understanding the drivers of stability across spatial scales is important for land management and policy decisions [25]. Stability in the availability of pollinators is also important from an agro-ecological resilience perspective as increased variation in animal pollination could reduce average yield and yield stability [51]. We further propose that the stability and ecological resilience brought about by enhancing the diversity of pollinator communities will contribute beyond agriculture and should be considered alongside longer term conservation targets focussed on maintaining and enhancing wider biodiversity.

Data accessibility. The data supporting the analyses are available from the University of Reading Research Data Archive http://dx.doi.org/10.17864/1947.291 [52].

Authors' contributions. D.S. collated datasets, analysed the data and wrote the manuscript based on initial ideas conceived by A.M.K. J.F. wrote

the R code for the initial data analyses, and along with M.A., M.P.D.G., D.K., B.J.P., S.G.P. and A.M.K. was involved in helping structure subsequent data analyses and in commenting on several early drafts of the manuscripts. B.B. produced the insect illustrations used in figure 1 in addition to contributing data. All other authors provided the data used in the analyses and contributed to revisions of the manuscript.

Competing interests. We declare we have no competing interests.

Funding. This study was supported EU COST Action Super-B project (STSM-FA1307-150416-070296) and D.S. by the University of Reading Research Endowment Trust Fund (E3530600) and NERC KE Fellowship NE/S006400/1. J.F. by DFG grant FR 3364/4-1; L.G.C. funded by FCT and EU project EUCLIPO-028360 and by CNPq 421668/2018-0; P.Q. 305157/2018-3; M.P.D.G. by an Insect Pollinators Initiative grant no. BB/I000348/1; D.K. by the Dutch Ministry of Agriculture, Nature and Food Quality (BO-11-011.01-011); A.J. and H.Z. by the Bee resources research funds (CAAS-ASTIP-IAR; NSFC31672500) in China; B.M.F. by a Productivity in Research Sponsorship (no. 308948/16-5), Brasilia-Brazil; M.M. and D.W. funded by Waitrose & Partners, Fruition PO and the University of Worcester; and C.W. funded by the DFG grant no. 405945293.

Acknowledgements. The authors wish to thank Riccardo Bommarco for his data contribution.

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
