## [Peer Review File · Proceedings of the Royal Society B: Biological Sciences]

Review History

RSPB-2020-1611.R0 (Original submission)

Review form: Reviewer 1

Recommendation

Reject – article is scientifically unsound

Scientific importance: Is the manuscript an original and important contribution to its field?

Good

General interest: Is the paper of sufficient general interest?

Good

Quality of the paper: Is the overall quality of the paper suitable?

Marginal

Is the length of the paper justified?

Yes

Should the paper be seen by a specialist statistical reviewer?

No

Do you have any concerns about statistical analyses in this paper? If so, please specify them explicitly in your report.

Yes

It is a condition of publication that authors make their supporting data, code and materials available - either as supplementary material or hosted in an external repository. Please rate, if applicable, the supporting data on the following criteria.

Is it accessible?

Yes

Is it clear?

Yes

Is it adequate?

Yes

Do you have any ethical concerns with this paper?

No

Comments to the Author

I found the topic extremely timely and interesting. Although I applaud the effort of putting together this interesting dataset, I have several major concerns on the analyses and therefore on the results of this study.

MAJOR COMMENTS

-Temporal stability analyses

The authors used three metrics of temporal stability: CV of total abundance, CV of species richness, average Bray-Curtis based on a minimum of 2 years of data to a maximum of 4-5 years. First, when estimating temporal stability with such short time-series there is a risk of biases emerging from different length of the time-series. A longer time series more likely include an extreme year that in turn increases temporal variation. This can be dangerous when comparing different regions as in this study. This requires a formal test to evaluate how much variation in stability can be explained by the number of years and if the tested explanatory variables are correlated with the length of the time series across studies. Another potential bias is related to the criterion d) for data inclusion (L.98): 500 m appears quite large. What is the average distance between fields for annual crops?

Another problem is related to Bray-Curtis. When pollinator communities exhibit low evenness (most ecological communities), by definition the Bray-Curtis is mathematically strongly dependent on the variation in the most abundant species. So there is a circular argument in testing this effect. Fig. 4 is what you always get if you have a dominant species in the community. If you still want to use a beta-diversity approach and keep your explanatory variables, it would be better to use a binary version of beta such as Baselga or Podani. This approach could test turnover in composition

I suggest the authors to read previous studies that aimed at disentangling the mechanisms of temporal stability papers (e.g. Lehman & Tilman, 2000; Loreau & De Mazancourt, 2008; Sasaki & Lauenroth 2010; Grman et al. 2010; Wilcox et al. 2017 and many others) to find inspiration an alternative and more elegant approaches to this problem. Quantifying the mechanisms of temporal stability in a community is not a trivial task.

-Shannon diversity as explanatory variables

First, the argument provided at line 153-154 to support the use of Shannon is partially incorrect. Shannon depends on species richness and evenness. This is why I suggest to remove this metric from the whole study and to use instead metrics whose ecological meaning can be univocally interpreted such as species richness. The large body of literature on stability has often tested species richness effect on stability. I am surprised that this was not done here.

-Modelling approach

The authors tested the effects of more than 10 variables in single mixed effect models using study ID as random effect. Some of these variables are continuous and some categorical. However, the authors tested for collinearity only for the continuous while potential collinearity involving categorical variables is disregarded. For instance, crop family, MFC, crop species are clearly dependent and cannot be tested together. The use of crop type as explanatory variable is not acceptable. You are testing a factor with 24 levels (correct?) in a model with already more than 10 other variables (See comment below). The chances of overfitting are very high. Model selection was performed using a stepwise approach that is extremely dangerous under these circumstances. Considering the potential problem related to model selection, this modelling approach is not acceptable. First, I suggest to consider only independent variables avoiding circular arguments (see above) that are related to clear ecological hypotheses (e.g. why was continent tested?). Second, you should test whether these variables (both continuous and categorical) are related to each other (e.g. using correlation, chi-squared, vifs etc.) and reduce the number of explanatory variables before building your model. Finally, to evaluate model selection uncertainty, I encourage the authors to present results from both full and reduced models besides a multi-model inference analysis. For the latter, you should present the list of plausible models (AICc difference below 7) with associated model weights and parameters.

-Introduction

Introduction: I would shorten the intro by removing/reducing some parts related to general trends and pollination (basically first three paragraphs can be condensed). I would instead focus the intro more on ecological theory related to community stability.

In conclusion, I see potential in the dataset but I find that the authors did not properly analyse the data and did not consider the large body of available research on community temporal stability. This is reflected in the bibliography where there are a very few methodological studies cited. This is a very interesting but complex area of research that needs a rigorous methodological approach. Besides the model selection problems outlined above, the study mixed very different questions: drivers of temporal stability vs. mechanisms of temporal stability. Most of the significant effects reported are simply telling us that there is a dominant species in most communities. Here, the authors really need to refocus their aims and hypotheses and select the appropriate methods to address those questions.

REFERENCES

- Grman, E., Lau, J. A., Schoolmaster Jr, D. R., & Gross, K. L. (2010). Mechanisms contributing to stability in ecosystem function depend on the environmental context. *Ecology letters*, 13(11), 1400-1410.
- Lehman, C. L., & Tilman, D. (2000). Biodiversity, stability, and productivity in competitive communities. *The American Naturalist*, 156(5), 534-552.
- Loreau, M. & De Mazancourt, C. (2008). Species synchrony and its drivers: neutral and nonneutral community dynamics in fluctuating environments. *Am. Nat.*, 172, E48-E66.
- Sasaki, T., & Lauenroth, W. K. (2011). Dominant species, rather than diversity, regulates temporal

stability of plant communities. *Oecologia*, 166(3), 761-768.

Wilcox, K. R., Tredennick, A. T., Koerner, S. E., Grman, E., Hallett, L. M., Avolio, M. L., ... & Alatalo, J. M. (2017). Asynchrony among local communities stabilises ecosystem function of metacommunities. *Ecology letters*, 20(12), 1534-1545.

And many others!

Review form: Reviewer 2 (Katherine Baldock)

Recommendation

Accept with minor revision (please list in comments)

Scientific importance: Is the manuscript an original and important contribution to its field?

Excellent

General interest: Is the paper of sufficient general interest?

Good

Quality of the paper: Is the overall quality of the paper suitable?

Excellent

Is the length of the paper justified?

Yes

Should the paper be seen by a specialist statistical reviewer?

No

Do you have any concerns about statistical analyses in this paper? If so, please specify them explicitly in your report.

Yes

It is a condition of publication that authors make their supporting data, code and materials available - either as supplementary material or hosted in an external repository. Please rate, if applicable, the supporting data on the following criteria.

Is it accessible?

N/A

Is it clear?

N/A

Is it adequate?

N/A

Do you have any ethical concerns with this paper?

No

Comments to the Author

This is an interesting study that brings together a large collection of datasets from across the globe to consider interannual variation in crop-visiting pollinator communities from multiple regions and climates, something that has only previously been considered before for single crop species. This is a worthwhile study which will add to the wider knowledge on pollinator services for crop pollination. The methods appear sound, although clarification on some details of the analyses would be beneficial. It would also be good to consider the wider implications of these

findings for crop production in a bit more detail in the Discussion section. And have the authors considered if proximity to semi-natural habitat might play an important role in the stability of pollinator communities between years? I have mainly minor comments on the manuscript as outlined below.

Methods

1. Regarding the equation in line 140, I am not exactly clear how the information obtained from the various visitation datasets was used in this calculation, and across which data the mean and SD that feed into this equation were calculated. Looking at Fig 2 it seems like CV was calculated for each crop at each study site, so were the mean and SD values used to calculate the CV values for each crop calculated across multiple sampling visits for each crop at each site? In some cases did this mean that the mean and SD were calculated across quite low numbers of datapoints? If so please state that the mean and SD were calculated from a small number of datapoints in some cases.
2. Line 159-160 – could a paired t-test have been used here if the total pollinator abundance and single/two-most, etc values came from the same crop/site?
3. Line 178 – were model residuals checked for heteroscedasticity?
4. Table S3 – what value is shown in this table – is this R? And what was the cut-off point you used to decide which variables were correlated? Also, please explain what CV_{top1Species} abundance and others mean in the legend.
5. Line 184. How was the most parsimonious model defined?

Results

6. Line 193 - What data were this mean and SD calculated across? The CV was calculated as the SD/mean across a particular dataset. Then it looks like the mean and SD are calculated here across a collection of CV values, but I'm not sure which ones, are there multiple CV values for each site, perhaps where more than two years were compared? In which case these mean and SD values may have been calculated across a small number of values (3 or 6). Are SD values higher for sites with more years of data?
7. Fig S1 – it will be difficult for colour-blind readers to pick out the different crop species, and it is not that easy for a non-colour blind person, although I appreciate you need a lot of colours here. Is there a way around this, e.g. put crop species on the x axis and then use a shorter code (1 upwards) to cross-reference to another table to indicate which study the dataset is from. As it is currently shown the reader needs to use a code to cross reference to another table anyway. Also, could you also add a symbol to labels on the x axis to make it easy to see which are MFC?
8. Table S4. Please can you add a column for crop species to this table so it is clearer which dominant species correspond with which crop. Pollinator species names need italicising.
9. Line 195 – do you have any comments on the results shown in Fig S3?
10. Line 199 – should the t statistics be reported as positive values?
11. Line 205 – can post hoc tests be used to show which crop species were significantly different from one another?
12. Could binomial models be used to test if the variables have an effect on whether there is or isn't a change in the dominant species between years? It would be interesting to know if this is more typical of certain crop types/climatic regions/flower types.

13. Table 2 talks about 'biome' but 'climatic region' as used elsewhere is a more appropriate term. It could also be interesting to look at the effect of biome, if there are enough datasets from different biomes.
14. Lines 217-219 – move this to the end of the paragraph.
15. Fig 4 could be included in Fig 3 as a fourth panel.
16. Fig 1 – another beetle taxa may be more appropriate for the picture as ladybirds aren't typically known as pollinators.

Discussion

17. Line 232 – 'richness and diversity' rather than pollinator communities as this variable didn't affect variation in total pollinator abundance.
18. Did you have any information on crop yields for any of the datasets and if so is it possible to examine how these varied among years and see if this was related to changes in pollinator abundance/richness/diversity?
19. Figs S2 and S3 – indicate any sig difs between temperate and tropical.
20. Line 253-254 – I suggest moving the honeybee text back into the main text as the discussions around the potential impact of managed pollinators on wild pollinators is important to consider in this context.
21. Line 262 – here do you mean stability of the crop pollinator community rather than the overall pollinator community? The stability of the crop pollinator community will also be influenced by the wider community of plant-pollinator interactions, so I'm not sure if you can really talk about the overall community stability while only considering visitors to one or a few plant species.
22. Is there semi-natural habitat around most of these crop species? The pollinators are likely to need more than just the crop species to forage on, especially for mass-flowering crop species, so semi-natural habitat and the plant species within these will also likely be important to community stability. It would be interesting to know if proximity to semi-natural habitat had an effect on the stability of crop visitation between years.
23. Line 294 – do you know which crop species had a bigger effect on changes in pollinator abundance? See earlier comment on post hoc tests.
24. Line 300 - could be worth highlighting that brinjal also known as aubergine or eggplant in other countries.
25. Para starting at line 318 - could this finding be due to different approaches to agricultural land management between temperate and tropical systems? Do all of the tropical systems in this study have lots of crops grown in a small area with large areas of individual crop species for the temperate systems?
26. Line 323 – greater number of crop or pollinator species?
27. Wider implications paragraph – I'm not sure if this is the best subheading for this section as currently written. There is also a change in referencing style at this point and several references are not in the reference list. I felt that this section could do with a bit more depth and specificity, for example (line 351) do you mean that *future* assessments of visitation rates or pollination services will help understanding the factors. Will lower variability in pollinator abundance/diversity/richness translate into higher pollination services? If it doesn't then

perhaps variation in pollinator community composition between years is not so important?

28. Line 354 – can you talk a bit more about how could pollinator communities be enhanced? And is proximity of semi-natural habitat and the complexity of the surrounding landscape going to play an important role here?

29. Check margins for Table S1.

Thank you for inviting me to review this manuscript, it is an interesting and thought-provoking study and clearly a lot of work has gone into assimilating the data used in this comprehensive meta-analysis.

Best wishes,
Katherine Baldock.

Decision letter (RSPB-2020-1611.R0)

06-Aug-2020

Dear Dr Senapathi:

I am writing to inform you that your manuscript RSPB-2020-1611 entitled "Wild insect diversity increases inter-annual stability in global crop pollinator communities" has, in its current form, been rejected for publication in Proceedings B.

This action has been taken on the advice of referees, who have recommended that substantial revisions are necessary. With this in mind we would be happy to consider a resubmission, provided the comments of the referees are fully addressed. However please note that this is not a provisional acceptance.

Sincerely,
 Dr Locke Rowe
 mailto: proceedingsb@royalsociety.org

Associate Editor

Board Member: 1

Comments to Author:

Your manuscript has been reviewed by two experts in the field. They, and I, agree that your dataset and the question you address are important and of value. However, both reviewers raise a number of issues, and most importantly one of the reviewers questions your entire statistical approach. As such, I cannot recommend your manuscript for publication. Please look carefully through their comments. Any revised version of this manuscript will need to address all of these, and I believe is going to require significant re-analyses of your dataset in line with the recommendations of this reviewer.

Reviewer(s)' Comments to Author:

Referee: 1

Comments to the Author(s)

I found the topic extremely timely and interesting. Although I applaud the effort of putting together this interesting dataset, I have several major concerns on the analyses and therefore on the results of this study.

MAJOR COMMENTS

-Temporal stability analyses

The authors used three metrics of temporal stability: CV of total abundance, CV of species richness, average Bray-Curtis based on a minimum of 2 years of data to a maximum of 4-5 years. First, when estimating temporal stability with such short time-series there is a risk of biases emerging from different length of the time-series. A longer time series more likely include an extreme year that in turn increases temporal variation. This can be dangerous when comparing different regions as in this study. This requires a formal test to evaluate how much variation in stability can be explained by the number of years and if the tested explanatory variables are correlated with the length of the time series across studies. Another potential bias is related to the criterion d) for data inclusion (L.98): 500 m appears quite large. What is the average distance between fields for annual crops?

Another problem is related to Bray-Curtis. When pollinator communities exhibit low evenness (most ecological communities), by definition the Bray-Curtis is mathematically strongly dependent on the variation in the most abundant species. So there is a circular argument in testing this effect. Fig. 4 is what you always get if you have a dominant species in the community. If you still want to use a beta-diversity approach and keep your explanatory variables, it would be better to use a binary version of beta such as Baselga or Podani. This approach could test turnover in composition

I suggest the authors to read previous studies that aimed at disentangling the mechanisms of temporal stability papers (e.g. Lehman & Tilman, 2000; Loreau & De Mazancourt, 2008; Sasaki & Lauenroth 2010; Grman et al. 2010; Wilcox et al. 2017 and many others) to find inspiration an alternative and more elegant approaches to this problem. Quantifying the mechanisms of temporal stability in a community is not a trivial task.

-Shannon diversity as explanatory variables

First, the argument provided at line 153-154 to support the use of Shannon is partially incorrect. Shannon depends on species richness and evenness. This is why I suggest to remove this metric from the whole study and to use instead metrics whose ecological meaning can be univocally interpreted such as species richness. The large body of literature on stability has often tested species richness effect on stability. I am surprised that this was not done here.

-Modelling approach

The authors tested the effects of more than 10 variables in single mixed effect models using study ID as random effect. Some of these variables are continuous and some categorical. However, the authors tested for collinearity only for the continuous while potential collinearity involving categorical variables is disregarded. For instance, crop family, MFC, crop species are clearly dependent and cannot be tested together. The use of crop type as explanatory variable is not acceptable. You are testing a factor with 24 levels (correct?) in a model with already more than 10 other variables (See comment below). The chances of overfitting are very high. Model selection was performed using a stepwise approach that is extremely dangerous under these circumstances. Considering the potential problem related to model selection, this modelling approach is not acceptable. First, I suggest to consider only independent variables avoiding circular arguments (see above) that are related to clear ecological hypotheses (e.g. why was continent tested?). Second, you should test whether these variables (both continuous and categorical) are related to each other (e.g. using correlation, chi-squared, vifs etc.) and reduce the number of explanatory variables before building your model. Finally, to evaluate model selection uncertainty, I encourage the authors to present results from both full and reduced models besides a multi-model inference analysis. For the latter, you should present the list of plausible models (AICc difference below 7) with associated model weights and parameters.

-Introduction

Introduction: I would shorten the intro by removing/reducing some parts related to general trends and pollination (basically first three paragraphs can be condensed). I would instead focus the intro more on ecological theory related to community stability.

In conclusion, I see potential in the dataset but I find that the authors did not properly analyse the data and did not consider the large body of available research on community temporal stability. This is reflected in the bibliography where there are a very few methodological studies cited. This is a very interesting but complex area of research that needs a rigorous methodological approach. Besides the model selection problems outlined above, the study mixed very different questions: drivers of temporal stability vs. mechanisms of temporal stability. Most of the significant effects reported are simply telling us that there is a dominant species in most communities. Here, the authors really need to refocus their aims and hypotheses and select the appropriate methods to address those questions.

REFERENCES

- Grman, E., Lau, J. A., Schoolmaster Jr, D. R., & Gross, K. L. (2010). Mechanisms contributing to stability in ecosystem function depend on the environmental context. *Ecology letters*, 13(11), 1400-1410.
- Lehman, C. L., & Tilman, D. (2000). Biodiversity, stability, and productivity in competitive communities. *The American Naturalist*, 156(5), 534-552.
- Loreau, M. & De Mazancourt, C. (2008). Species synchrony and its drivers: neutral and nonneutral community dynamics in fluctuating environments. *Am. Nat.*, 172, E48-E66.
- Sasaki, T., & Lauenroth, W. K. (2011). Dominant species, rather than diversity, regulates temporal stability of plant communities. *Oecologia*, 166(3), 761-768.
- Wilcox, K. R., Tredennick, A. T., Koerner, S. E., Grman, E., Hallett, L. M., Avolio, M. L., ... & Alatalo, J. M. (2017). Asynchrony among local communities stabilises ecosystem function of metacommunities. *Ecology letters*, 20(12), 1534-1545.

And many others!

Referee: 2

Comments to the Author(s)

This is an interesting study that brings together a large collection of datasets from across the globe to consider interannual variation in crop-visiting pollinator communities from multiple

regions and climates, something that has only previously been considered before for single crop species. This is a worthwhile study which will add to the wider knowledge on pollinator services for crop pollination. The methods appear sound, although clarification on some details of the analyses would be beneficial. It would also be good to consider the wider implications of these findings for crop production in a bit more detail in the Discussion section. And have the authors considered if proximity to semi-natural habitat might play an important role in the stability of pollinator communities between years? I have mainly minor comments on the manuscript as outlined below.

Methods

1. Regarding the equation in line 140, I am not exactly clear how the information obtained from the various visitation datasets was used in this calculation, and across which data the mean and SD that feed into this equation were calculated. Looking at Fig 2 it seems like CV was calculated for each crop at each study site, so were the mean and SD values used to calculate the CV values for each crop calculated across multiple sampling visits for each crop at each site? In some cases did this mean that the mean and SD were calculated across quite low numbers of datapoints? If so please state that the mean and SD were calculated from a small number of datapoints in some cases.
2. Line 159-160 – could a paired t-test have been used here if the total pollinator abundance and single/two-most, etc values came from the same crop/site?
3. Line 178 – were model residuals checked for heteroscedasticity?
4. Table S3 – what value is shown in this table – is this R? And what was the cut-off point you used to decide which variables were correlated? Also, please explain what CV_{top1Species} abundance and others mean in the legend.
5. Line 184. How was the most parsimonious model defined?

Results

6. Line 193 - What data were this mean and SD calculated across? The CV was calculated as the SD/mean across a particular dataset. Then it looks like the mean and SD are calculated here across a collection of CV values, but I'm not sure which ones, are there multiple CV values for each site, perhaps where more than two years were compared? In which case these mean and SD values may have been calculated across a small number of values (3 or 6). Are SD values higher for sites with more years of data?
7. Fig S1 – it will be difficult for colour-blind readers to pick out the different crop species, and it is not that easy for a non-colour blind person, although I appreciate you need a lot of colours here. Is there a way around this, e.g. put crop species on the x axis and then use a shorter code (1 upwards) to cross-reference to another table to indicate which study the dataset is from. As it is currently shown the reader needs to use a code to cross reference to another table anyway. Also, could you also add a symbol to labels on the x axis to make it easy to see which are MFC?
8. Table S4. Please can you add a column for crop species to this table so it is clearer which dominant species correspond with which crop. Pollinator species names need italicising.
9. Line 195 – do you have any comments on the results shown in Fig S3?
10. Line 199 – should the t statistics be reported as positive values?
11. Line 205 – can post hoc tests be used to show which crop species were significantly different from one another?

12. Could binomial models be used to test if the variables have an effect on whether there is or isn't a change in the dominant species between years? It would be interesting to know if this is more typical of certain crop types/climatic regions/flower types.
13. Table 2 talks about 'biome' but 'climatic region' as used elsewhere is a more appropriate term. It could also be interesting to look at the effect of biome, if there are enough datasets from different biomes.
14. Lines 217-219 - move this to the end of the paragraph.
15. Fig 4 could be included in Fig 3 as a fourth panel.
16. Fig 1 - another beetle taxa may be more appropriate for the picture as ladybirds aren't typically known as pollinators.
- Discussion
17. Line 232 - 'richness and diversity' rather than pollinator communities as this variable didn't affect variation in total pollinator abundance.
18. Did you have any information on crop yields for any of the datasets and if so is it possible to examine how these varied among years and see if this was related to changes in pollinator abundance/richness/diversity?
19. Figs S2 and S3 - indicate any sig difs between temperate and tropical.
20. Line 253-254 - I suggest moving the honeybee text back into the main text as the discussions around the potential impact of managed pollinators on wild pollinators is important to consider in this context.
21. Line 262 - here do you mean stability of the crop pollinator community rather than the overall pollinator community? The stability of the crop pollinator community will also be influenced by the wider community of plant-pollinator interactions, so I'm not sure if you can really talk about the overall community stability while only considering visitors to one or a few plant species.
22. Is there semi-natural habitat around most of these crop species? The pollinators are likely to need more than just the crop species to forage on, especially for mass-flowering crop species, so semi-natural habitat and the plant species within these will also likely be important to community stability. It would be interesting to know if proximity to semi-natural habitat had an effect on the stability of crop visitation between years.
23. Line 294 - do you know which crop species had a bigger effect on changes in pollinator abundance? See earlier comment on post hoc tests.
24. Line 300 - could be worth highlighting that brinjal also known as aubergine or eggplant in other countries.
25. Para starting at line 318 - could this finding be due to different approaches to agricultural land management between temperate and tropical systems? Do all of the tropical systems in this study have lots of crops grown in a small area with large areas of individual crop species for the temperate systems?
26. Line 323 - greater number of crop or pollinator species?
27. Wider implications paragraph - I'm not sure if this is the best subheading for this section as currently written. There is also a change in referencing style at this point and several references

are not in the reference list. I felt that this section could do with a bit more depth and specificity, for example (line 351) do you mean that *future* assessments of visitation rates or pollination services will help understanding the factors. Will lower variability in pollinator abundance/diversity/richness translate into higher pollination services? If it doesn't then perhaps variation in pollinator community composition between years is not so important?

28. Line 354 – can you talk a bit more about how could pollinator communities be enhanced? And is proximity of semi-natural habitat and the complexity of the surrounding landscape going to play an important role here?

29. Check margins for Table S1.

Thank you for inviting me to review this manuscript, it is an interesting and thought-provoking study and clearly a lot of work has gone into assimilating the data used in this comprehensive meta-analysis.

Best wishes,
Katherine Baldock.

Author's Response to Decision Letter for (RSPB-2020-1611.R0)

See Appendix A.

RSPB-2021-0212.R0

Review form: Reviewer 1

Recommendation

Accept with minor revision (please list in comments)

Scientific importance: Is the manuscript an original and important contribution to its field?

Good

General interest: Is the paper of sufficient general interest?

Excellent

Quality of the paper: Is the overall quality of the paper suitable?

Good

Is the length of the paper justified?

Yes

Should the paper be seen by a specialist statistical reviewer?

No

Do you have any concerns about statistical analyses in this paper? If so, please specify them explicitly in your report.

No

It is a condition of publication that authors make their supporting data, code and materials available - either as supplementary material or hosted in an external repository. Please rate, if applicable, the supporting data on the following criteria.

Is it accessible?

No

Is it clear?

Yes

Is it adequate?

No

Do you have any ethical concerns with this paper?

No

Comments to the Author

I find the revised version much improved and I am happy to see that most of my suggestions have been carefully addressed. I have only minor comments and a potential major suggestion that I missed from the first round of review (sorry!).

Starting from the latter, by reading L. 272-286 I was wondering if you could test the effect on stability of having the honeybee as the dominant species. You could classify the studies in two categories according to the identity of the dominant species (honeybee vs. wild bees). I imagine that for the same study you might have years where you have different dominant species but there should be a clever way to split studies to test the species identity effect. This could be a nice add-on.

Minor comments:

L.151 Here, I see that crop species is still included in the model. However, in the response letter the authors stated that they have excluded this variable from the modelling. I had suggested to omit this variable as crop species has too many levels and practically overlaps with study ID (your random). Please clarify.

L.164-165 Please state explicitly that you used ML and not REML to estimate your models in the multi-model inference analysis.

L. 268-270 Please remove speculations about the effect of landscape heterogeneity on stability. This statement is not supported by your data.

Decision letter (RSPB-2021-0212.R0)

15-Feb-2021

Dear Dr Senapathi:

Your manuscript has now been peer reviewed and the reviews have been assessed by an Associate Editor. The reviewers' comments (not including confidential comments to the Editor) and the comments from the Associate Editor are included at the end of this email for your reference. As you will see, the reviewers and the Editors have raised some concerns with your manuscript and we would like to invite you to revise your manuscript to address them.

Research ethics:

Use of animals and field studies:

It is a condition of publication that you make available the data and research materials supporting the results in the article (<https://royalsociety.org/journals/authors/author-guidelines/#data>). Datasets should be deposited in an appropriate publicly available repository and details of the associated accession number, link or DOI to the datasets must be included in the Data Accessibility section of the article (<https://royalsociety.org/journals/ethics-policies/data-sharing-mining/>). Reference(s) to datasets should also be included in the reference list of the article with DOIs (where available).

Please submit a copy of your revised paper within three weeks. If we do not hear from you within this time your manuscript will be rejected. If you are unable to meet this deadline please let us know as soon as possible, as we may be able to grant a short extension.

Best wishes,
Dr Locke Rowe
mailto:proceedingsb@royalsociety.org

Associate Editor Board Member

Comments to Author:

Your revised manuscript has been received very positively by the reviewer, and so I only have a few comments:

- 1) you must make your dataset completely and clearly available by depositing the full dataset in Dryad, or a similar server. A partial list of links to single datasets is not appropriate
- 2) the reviewer requests a stability analysis. I think this would be a nice add on and increase the value of your paper, but also recognise that it could be considerable work. I leave it up to you as to whether you would like to include this or not.
- 3) explain the issue with 'crop species' as identified by the reviewer
- 4) make the model statement as requested
- 5) make amendments around the section on heterogeneity/stability as requested

Reviewer(s)' Comments to Author:

Referee: 1

Comments to the Author(s).

I find the revised version much improved and I am happy to see that most of my suggestions have been carefully addressed. I have only minor comments and a potential major suggestion that I missed from the first round of review (sorry!).

Starting from the latter, by reading L. 272-286 I was wondering if you could test the effect on stability of having the honeybee as the dominant species. You could classify the studies in two categories according to the identity of the dominant species (honeybee vs. wild bees). I imagine that for the same study you might have years where you have different dominant species but there should be a clever way to split studies to test the species identity effect. This could be a nice add-on.

Minor comments:

L.151 Here, I see that crop species is still included in the model. However, in the response letter the authors stated that they have excluded this variable from the modelling. I had suggested to omit this variable as crop species has too many levels and practically overlaps with study ID (your random). Please clarify.

L.164-165 Please state explicitly that you used ML and not REML to estimate your models in the multi-model inference analysis.

L. 268-270 Please remove speculations about the effect of landscape heterogeneity on stability. This statement is not supported by your data.

Author's Response to Decision Letter for (RSPB-2021-0212.R0)

See Appendix B.

Decision letter (RSPB-2021-0212.R1)

23-Feb-2021

Dear Dr Senapathi

I am pleased to inform you that your manuscript entitled "Wild insect diversity increases inter-annual stability in global crop pollinator communities" has been accepted for publication in Proceedings B.

Open Access

Paper charges

Sincerely,

Dr Locke Rowe

Associate Editor:

Board Member

Comments to Author:

Thank you for the revised version and associated response to reviewers. It's great to see the stability analysis included, and I'm delighted to be able to recommend your manuscript for acceptance. Thank you for submitting this work to us, and for engaging so positively throughout with the review process.

Appendix A

The Editor,
Proceedings of the Royal Society
B: Biological Sciences

Dr. Deepa Senapathi
Senior Research Fellow

Centre for Agri-Environmental Research (CAER)
School of Agriculture, Policy & Development
Earley Gate, Whiteknights Campus
University of Reading
Reading RG6 6AR

Phone: +44 (0)118 378 4541

Email: g.d.senapathi@reading.ac.uk

Website:

<https://www.reading.ac.uk/apd/staff/g-d-senapathi.aspx>

26 January 2021

Dear Editor,

Subject: Resubmission of research article to Proceedings of the Royal Society B

We enclose a revised manuscript entitled: “*Wild insect diversity increases inter-annual stability in global crop pollinator communities*” by Senapathi et al. for resubmission to Proceedings of the Royal Society B.

We thank the Associate Editor and the two reviewers for their valuable comments and suggestions. We have taken on board their opinions and have made significant changes to the model selection methodology as well as included some additional analyses to check the sensitivity and robustness of our findings. Our responses to each of the specific comments raised by the reviewers are detailed below this covering letter.

Multi-year studies on crop pollinators are rare and our study is novel in its approach to characterise temporal patterns in crop pollinator communities worldwide using studies focussed on both annual and perennial crops. Using 43 datasets from 12 countries across six continents on 21 crop species, we show that higher pollinator diversity confers greater inter-annual stability in crop pollinator communities but also that dominant species play a key role alongside diversity in conferring stability of pollinator communities. To the best of our knowledge, our study is the first to explore temporal variation in pollinator communities across different crops.

I hope you will find that this work is both of sufficient scientific merit and of general interest to be considered for publication in your journal. This work has not been published, nor is it under consideration for publication elsewhere.

Yours sincerely,

Dr Deepa Senapathi

Responses to reviewer comments

Reviewer(s)' Comments to Author:

Referee: 1

Comments to the Author(s)

I found the topic extremely timely and interesting. Although I applaud the effort of putting together this interesting dataset, I have several major concerns on the analyses and therefore on the results of this study.

MAJOR COMMENTS

-Temporal stability analyses

The authors used three metrics of temporal stability: CV of total abundance, CV of species richness, average Bray-Curtis based on a minimum of 2 years of data to a maximum of 4-5 years. First, when estimating temporal stability with such short time-series there is a risk of biases emerging from different length of the time-series. A longer time series more likely include an extreme year that in turn increases temporal variation. This can be dangerous when comparing different regions as in this study. This requires a formal test to evaluate how much variation in stability can be explained by the number of years and if the tested explanatory variables are correlated with the length of the time series across studies.

The authors agree with the reviewer and we now use CV of richness and abundance for the whole study period but also calculated the CV for every pairwise year comparisons (i.e. Y1&Y2; Y2&Y3; y3&Y4 etc). The models run with these pairwise CVs have been included in our revised analyses (please see lines 132-134 &153-155 in the methods section and also Table 2) in the main manuscript, alongside the models that consider the overall CV across all years of the studies.

A correlation test between the number of years of the studies and response variables showed no significant effect on CV of abundance ($t = 0.92$, $df = 372$, $p\text{-value} = 0.36$) or CV of richness ($t = 0.16$, $df = 367$, $p\text{-value} = 0.87$)

Another potential bias is related to the criterion d for data inclusion (L.98): 500 m appears quite large. What is the average distance between fields for annual crops?

The 500m limit was chosen not as an average distance between annual crop fields but as an indicator of flight distance of both solitary and social crop pollinators (for e.g. Wolf and Moritz 2008; Hofmann et al. 2020). The average distance between fields for annual crops varied greatly between geographic regions. The 500m maximum limit accounts for the foraging distance of pollinator communities that would enable them to visit the same crops grown in different fields in different years. While perennial crop fields remained the same between years, the minimum recorded distance between fields in our annual crops was 50m and the maximum was set at 500m for the reasons outlined above.

Another problem is related to Bray-Curtis. When pollinator communities exhibit low evenness (most ecological communities), by definition the Bray-Curtis is mathematically strongly dependent on the variation in the most abundant species. So there is a circular argument in testing this effect. Fig. 4 is what you always get if you have a dominant species in the community. If you still want to use a beta-diversity approach and keep your explanatory variables, it would be better to use a binary version of beta such as Baselga or Podani. This approach could test turnover in composition.

We agree with your point of the circular argument pertaining to the Bray Curtis and dominant species relationship. We have therefore excluded the Bray Curtis analyses and any associated text in the revised manuscript. Thank you for this good suggestion.

I suggest the authors to read previous studies that aimed at disentangling the mechanisms of temporal stability papers (e.g. Lehman & Tilman , 2000; Loreau & De Mazancourt, 2008; Sasaki & Lauenroth 2010; Grman ety al. 2010; Wilcox et al. 2017 and many others) to find inspiration an alternative and more elegant approaches to this problem. Quantifying the mechanisms of temporal stability in a community is not a trivial task.

We have read the suggested studies and incorporated some of them in the revised manuscript (in the introduction, please see lines 50-60). The analyses have also been revised to incorporate a better model selection approach as suggested (please see lines 160-165). A series of candidate models were constructed for each response variable that ensured that correlated variables were not included in the same models. Each candidate model was ‘dredged’ to obtain a series of plausible intermediate models. Intermediate models with Δ AICc value < 7 of the model with lowest AICc were averaged to obtain the final outputs and this detailed in the main manuscript.

-Shannon diversity as explanatory variables

First, the argument provided at line 153-154 to support the use of Shannon is partially incorrect. Shannon depends on species richness and evenness. This is why I suggest to remove this metric from the whole study and to use instead metrics whose ecological meaning can be univocally interpreted such as species richness. The large body of literature on stability has often tested species richness effect on stability. I am surprised that this was not done here.

We do take the reviewer’s point about choosing the right metrics. The Shannon diversity index was chosen as Shannon H is simply the log-transformed “effective number of species = true diversity”. It accounts for evenness of the species present, thus reflecting effective diversity, and also is less sensitive to sampling effects than species richness (Krebs 1999; Jost 2006).

-Modelling approach

The authors tested the effects of more than 10 variables in single mixed effect models using study ID as random effect. Some of these variables are continuous and some categorical. However, the authors tested for collinearity only for the continuous while potential collinearity involving categorical variables is disregarded. For instance, crop family, MFC, crop species are clearly dependent and cannot be tested together. The use of crop type as explanatory variable is not acceptable. You are testing a factor with 24 levels (correct?) in a model with already more than 10 other variables (See comment below). The chances of overfitting are very high.

We tested for collinearity of both continuous and categorical variables and we apologize that this was not clearly described in the methods as it was written. For example, collinearity between MFC and crop was tested and since a significant effect was found, MFC and crop were not included in the same models. In our original analyses, a series of candidate models were run for each response variable and the results presented showed the significant effects of the best reduced model. However, please note that as per your suggestion, the entire modelling approach has been revised (details below and in lines 160-165 of the main manuscript).

Model selection was performed using a stepwise approach that is extremely dangerous under these circumstances. Considering the potential problem related to model selection, this modelling approach is not acceptable. First, I suggest to consider only independent variables avoiding circular arguments (see above) that are related to clear ecological hypotheses (e.g. why was continent tested?). Second, you should test whether these variables (both continuous and categorical) are related to each other (e.g. using correlation, chi-squared, vifs etc.) and reduce the number of explanatory variables before building your model. Finally, to evaluate model selection uncertainty, I encourage the authors to present results from both full and reduced models besides a multi-model inference analysis. For the latter, you should present the list of plausible models (AICc difference below 7) with associated model weights and parameters.

The entire modelling approach has been revised – for each response variable (i.e. CV abundance and CV richness), a set of candidate models were set up (ensuring that collinear variables, continuous or categorical were not included in the same model). These models were then dredged using the MuMIn package in R, and model averaging of output models where AICc difference was below 7 performed (as suggested by the reviewer). The methods are detailed in lines 160-165 of the main manuscript. The results of the model averaging are now provided as a revised Table 2 in the main manuscript and in supplementary Table S5.

Introduction: I would shorten the intro by removing/reducing some parts related to general trends and pollination (basically first three paragraphs can be condensed). I would instead focus the intro more on ecological theory related to community stability.

We agree and shortened the general text on crop pollination to include studies on the theory of ecological stability (please see lines 50-60). While doing this we kept the focus on the crop pollinator stability and the factors that influence it, rather than contributing in long detail about ecological theory on community stability.

In conclusion, I see potential in the dataset but I find that the authors did not properly analyse the data and did not consider the large body of available research on community temporal stability. This is reflected in the bibliography where there are a very few methodological studies cited. This is a very interesting but complex area of research that needs a rigorous methodological approach. Besides the model selection problems outlined above, the study mixed very different questions: drivers of temporal stability vs. mechanisms of temporal stability. Most of the significant effects reported are simply telling us that there is a dominant species in most communities. Here, the authors really need to refocus their aims and hypotheses and select the appropriate methods to address those questions.

We thank the reviewer for their valuable suggestions and their comments. A new analysis looking at the effect of dominance species on asynchronous and synchronous pollinator communities has also been included to try and understand the mechanisms of temporal stability (please see lines 178-189 for methods and lines 235-243 for results and new Figure 4). We hope that the revised model selection method as well as the additional analyses to test for the robustness and sensitivity of our results contribute towards addressing the reviewer's concerns.

REFERENCES

- Grman, E., Lau, J. A., Schoolmaster Jr, D. R., & Gross, K. L. (2010). Mechanisms contributing to stability in ecosystem function depend on the environmental context. Ecology letters, 13(11), 1400-1410.*
- Lehman, C. L., & Tilman, D. (2000). Biodiversity, stability, and productivity in competitive communities. The American Naturalist, 156(5), 534-552.*
- Loreau, M. & De Mazancourt, C. (2008). Species synchrony and its drivers: neutral and nonneutral community dynamics in fluctuating environments. Am. Nat., 172, E48–E66.*
- Sasaki, T., & Lauenroth, W. K. (2011). Dominant species, rather than diversity, regulates temporal stability of plant communities. Oecologia, 166(3), 761-768.*
- Wilcox, K. R., Tredennick, A. T., Koerner, S. E., Grman, E., Hallett, L. M., Avolio, M. L., ... & Alatalo, J. M. (2017). Asynchrony among local communities stabilises ecosystem function of metacommunities. Ecology letters, 20(12), 1534-1545.*

And many others!

We thank the reviewer for the suggested references and have incorporated them in the introduction (see lines 50-60) and also where relevant in other parts of the manuscript.

Referee: 2

Comments to the Author(s)

This is an interesting study that brings together a large collection of datasets from across the globe to consider interannual variation in crop-visiting pollinator communities from multiple regions and climates, something that has only previously been considered before for single crop species. This is a worthwhile study which will add to the wider knowledge on pollinator services for crop pollination. The methods appear sound, although clarification on some details of the analyses would be beneficial. It would also be good to consider the wider implications of these findings for crop production in a bit more detail in the Discussion section. And have the authors considered if proximity to semi-natural habitat might play an important role in the stability of pollinator communities between years? I have mainly minor comments on the manuscript as outlined below.

Methods

1. Regarding the equation in line 140, I am not exactly clear how the information obtained from the various visitation datasets was used in this calculation, and across which data the mean and SD that feed into this equation were calculated. Looking at Fig 2 it seems like CV was calculated for each crop at each study site, so were the mean and SD values used to calculate the CV values for each crop calculated across multiple sampling visits for each crop at each site? In some cases did this mean that the mean and SD were calculated across quite low numbers of datapoints? If so please state that the mean and SD were calculated from a small number of datapoints in some cases.

We apologise that this was not well explained in the previous version. CV of pollinator abundance and CV of pollinator species richness was calculated for each site within each study. Since each study had multiple sites then an overall mean CV and SD could be calculated for each study utilising all the CV values of the sites within each study or for each crop (please see lines 197-199). The SD provided in supplementary figure S1 reflects the variation across sites within a dataset rather than a small number of datapoints. Table S1 provides the number of sites in each study dataset which have been used for these calculations. However, as the new analyses no longer show significance for crop species, this sub-section of Figure 2 has been removed.

2. Line 159-160 – could a paired t-test have been used here if the total pollinator abundance and single/two-most, etc values came from the same crop/site?

Thanks for suggesting this. We now used paired t-tests and the significance of the relationships remain unchanged (please see lines 167-176 for methods & lines 227-233 for results).

3. Line 178 – were model residuals checked for heteroscedasticity?

Yes, the models were checked for heteroscedasticity and we have now mentioned this in line 163 of the main manuscript.

4. Table S3 – what value is shown in this table – is this R? And what was the cut-off point you used to decide which variables were correlated? Also, please explain what CVtop1Species abundance and others mean in the legend.

We've provided Pearson's correlation co-efficient in Table S3. We have revised the table to indicate which variables were significantly correlated. CVtop1species abundance refers to the CV of abundance of the single most dominant species. This and other terms have now been clarified in the table and legend.

5. Line 184. How was the most parsimonious model defined?

Modelling approach has been overhauled (please see lines 160-165 in the methods). A series of candidate models have now been set up, dredged, and models with <7 AICc difference averaged (as suggested by reviewer 1) to provide the final results – please see table 2 in the main manuscript and table S5 in the supplementary material.

Results

6. Line 193 - What data were this mean and SD calculated across? The CV was calculated as the SD/mean across a particular dataset. Then it looks like the mean and SD are calculated here across a collection of CV values, but I'm not sure which ones, are there multiple CV values for each site, perhaps where more than two years were compared? In which case these mean and SD values may have been calculated across a small number of values (3 or 6). Are SD values higher for sites with more years of data?

Thank for you highlighting parts of the manuscript where we needed to improve clarity. The CV of pollinator abundance and CV of pollinator species richness was calculated for each site within each study/dataset. Each site has a single CV value calculated across all the years of data. Since each study had multiple sites then an overall mean CV and SD could be calculated for each study utilising all the CV values of the sites within each study (please see lines 197-199). Table S1 provides the number of sites in each study dataset which have been used for these calculations. There is no indication that SD values are higher for studies with more years of data.

7. Fig S1 – it will be difficult for colour-blind readers to pick out the different crop species, and it is not that easy for a non-colour blind person, although I appreciate you need a lot of colours here. Is there a way around this, e.g. put crop species on the x axis and then use a shorter code (1 upwards) to cross-reference to another table to indicate which study the dataset is from. As it is currently shown the reader needs to use a code to cross reference to another table anyway. Also, could you also add a symbol to labels on the x axis to make it easy to see which are MFC?

We acknowledge that the previous version of this figure was not easy to distinguish between the different crop types. Fig S1 has been revised to show brighter colours with increased line width. It still provides the Study ID on the x axis but more distinct colours have been used to represent the different crops and MFC have been indicated with an * in the legend

8. Table S4. Please can you add a column for crop species to this table so it is clearer which dominant species correspond with which crop. Pollinator species names need italicising.

We agree and thank you for pointing out these minor but important shortcomings. Crop species column has been added, and pollinator names are now italicised.

9. Line 195 – do you have any comments on the results shown in Fig S3?

Please see lines 326-340 in the discussion section where this effect is now elaborated on and discussed

10. Line 199 – should the t statistics be reported as positive values?

The negative or positive values of the t statistics were dependent on the order in which the variables were compared. The t statistic has now been reported as positive values (see lines 228-233)

11. Line 205 – can post hoc tests be used to show which crop species were significantly different from one another?

The crop species effect does not show up as significant in the new modelling approach. Also, considering we have several studies that look at same crops in different regions while other studies focus on a single crop species it would be difficult to disentangle the differences that can be specifically attributed to the crop itself. To answer this question in a robust manner we would need multiple studies of one crop from a single region, which we currently do not have.

12. Could binomial models be used to test if the variables have an effect on whether there is or isn't a change in the dominant species between years? It would be interesting to know if this is more typical of certain crop types/climatic regions/flower types.

We apologise but are unsure of what the reviewer is suggesting here. If the reviewer is asking for an analysis of effects of landscape and time on dominance (dominant vs non dominant in a binary way) of individual species, we agree that such analyses could bring up interesting results, if followed by a subsequent trait analyses but would divert from the main focus of this manuscript. We acknowledge the potential value of such analyses but suggest that they would be better suited for a future manuscript as they are out of the scope of this paper.

13. Table 2 talks about 'biome' but 'climatic region' as used elsewhere is a more appropriate term. It could also be interesting to look at the effect of biome, if there are enough datasets from different biomes.

This is a good point but as there are not enough datasets in this study that cover all the different biomes, so we have only used climatic region and have now ensured consistency in terminology across the manuscript

14. Lines 217-219 – move this to the end of the paragraph.

Moved to the end of the paragraph as suggested

15. Fig 4 could be included in Fig 3 as a fourth panel.

Original Figure 4 has been removed as we are no longer including Bray Curtis analyses in the manuscript but replace by a different figure showing the impact of dominant species on overall abundance synchronous and asynchronous communities

16. Fig 1 – another beetle taxa may be more appropriate for the picture as ladybirds aren't typically known as pollinators.

The ladybird has been replaced by another beetle taxa as suggested.

Discussion

17. Line 232 – 'richness and diversity' rather than pollinator communities as this variable didn't affect variation in total pollinator abundance.

Our models show that when the dominant species remains the same between years there is less variation in pollinator abundance. Therefore, we have modified this sentence as

follows – “the variation observed in pollinator communities is also driven by the dominant species changes across the years” (see lines 250-251).

18. Did you have any information on crop yields for any of the datasets and if so is it possible to examine how these varied among years and see if this was related to changes in pollinator abundance/richness/diversity?

We may consider this for future analyses when more studies are available. Currently there are not enough datasets within this study to analyse the relationships with yield, but this is a good point for future research.

19. Figs S2 and S3 – indicate any sig difs between temperate and tropical.

We have now indicated significance values in the figures. We also discuss these two effects in lines 328-330 and lines 336-340 of the discussion respectively.

20. Line 253-254 – I suggest moving the honeybee text back into the main text as the discussions around the potential impact of managed pollinators on wild pollinators is important to consider in this context.

We agree and now moved the honeybee text to the main text in the discussion section (please see line 288-296)

21. Line 262 – here do you mean stability of the crop pollinator community rather than the overall pollinator community? The stability of the crop pollinator community will also be influenced by the wider community of plant-pollinator interactions, so I'm not sure if you can really talk about the overall community stability while only considering visitors to one or a few plant species.

Yes, we mean the crop pollinator community, and this has been clarified in the text now. Thanks for pointing this out, please see line 282.

22. Is there semi-natural habitat around most of these crop species? The pollinators are likely to need more than just the crop species to forage on, especially for mass-flowering crop species, so semi-natural habitat and the plant species within these will also likely be important to community stability. It would be interesting to know if proximity to semi-natural habitat had an effect on the stability of crop visitation between years.

This is an interesting suggestion, but as not all the studies have this information, we decided to suggest this as questions for future research with more studies included that will allow for robust landscape analyses.

23. Line 294 – do you know which crop species had a bigger effect on changes in pollinator abundance? See earlier comment on post hoc tests.

Please see response above as to why we did not test for crop species effect. Crop species does not show up as significant in the revised analyses and the fact that multiple studies from several different regions focussed on the same crops while others only focussed on a single crop species make these effects impossible to disentangle.

24. Line 300 - could be worth highlighting that brinjal also known as aubergine or eggplant in other countries.

This has now been mentioned in the manuscript in the methods section (line 109) describing our datasets

25. Para starting at line 318 - could this finding be due to different approaches to agricultural land management between temperate and tropical systems? Do all of the tropical systems in this study have lots of crops grown in a small area with large areas of individual crop species for the temperate systems?

Yes, this is a possibility. While we cannot state that ALL tropical cropping conforms to the multi-crop system, the majority of the tropical studies from India do. The exceptions are the cotton studies from Southern US but since the crop species effect is not significant in the new models we have not elaborated on these points in the revised discussion.

26. Line 323 – greater number of crop or pollinator species?

We apologise for the ambiguity. We are referring to the pollinator community here, and the text has been revised to reflect this (please see line 308)

*27. Wider implications paragraph – I'm not sure if this is the best subheading for this section as currently written. There is also a change in referencing style at this point and several references are not in the reference list. I felt that this section could do with a bit more depth and specificity, for example (line 351) do you mean that *future* assessments of visitation rates or pollination services will help understanding the factors. Will lower variability in pollinator abundance/diversity/richness translate into higher pollination services? If it doesn't then perhaps variation in pollinator community composition between years is not so important?*

This section has been revised and re-written to address your points, please see line 356-369. We have also deleted the sub-heading of 'wider implications' and ensured that the references are consistent with the rest of the text.

28. Line 354 – can you talk a bit more about how could pollinator communities be

enhanced? And is proximity of semi-natural habitat and the complexity of the surrounding landscape going to play an important role here?

We do not have the surrounding landscape data for our studies to be able to analyse or infer the impact of landscape on the pollinator communities within this set of studies. We do discuss the importance of diversity as well as the role played by dominant species (including potentially managed pollinators) in providing stability but do not explicitly discuss pollinator enhancement because we could not test for their effects on pollinator stability. We do mention this topic as something to be considered in both research and practice in our final paragraph (lines 366-369).

29. Check margins for Table S1.

Checked and adjusted

Thank you for inviting me to review this manuscript, it is an interesting and thought-provoking study and clearly a lot of work has gone into assimilating the data used in this comprehensive meta-analysis.

Best wishes,

Katherine Baldock.

Thank you very much for taking the time to revise our manuscript. Your comments have greatly improved our manuscript.

References cited in responses to reviewers document

- Hofmann, M. M., A. Fleischmann, et al. (2020). "Foraging distances in six species of solitary bees with body lengths of 6 to 15 mm, inferred from individual tagging, suggest 150 m-rule-of-thumb for flower strip distances." Journal of Hymenoptera Research **77**.
- Jost, L. (2006). "Entropy and diversity." Oikos **113**(2): 363-375.
- Krebs, C. J. (1999). Ecological Methodology, Addison-Wesley Educational Publishers, Inc.
- Wolf, S. and R. F. A. Moritz (2008). "Foraging distance in *Bombus terrestris* L. (Hymenoptera: Apidae)." Apidologie **39**(4): 419-427.

Appendix B

Dear Editor,

Subject: Response to Associate Editor and reviewer comments and revised article with track changes

We enclose a revised manuscript entitled: “*Wild insect diversity increases inter-annual stability in global crop pollinator communities*” by Senapathi et al. for consideration of publication in the Proceedings of the Royal Society B.

We thank the Associate Editor and the reviewer for their valuable comments and suggestions. Our responses to each of the specific comments raised are detailed below. We also include, as requested, a copy of the manuscript with track changes, attached below the specific response to the reviewers.

I am corresponding on behalf of all the co-authors and look forward to receiving your response.

Thanks again.

Sincerely,

Deepa Senapathi

Responses to specific comments

Associate Editor Board Member

Comments to Author:

Your revised manuscript has been received very positively by the reviewer, and so I only have a few comments:

1) you must make your dataset completely and clearly available by depositing the full dataset in Dryad, or a similar server. A partial list of links to single datasets is not appropriate

Thank you for clarifying the format in which this data needs to be deposited. I have now deposited the final data files used in this manuscript in a single repository held by the University of Reading. The data is open access under a creative commons license and can be accessed at the following DOI <http://dx.doi.org/10.17864/1947.291>. A data accessibility has been statement added to the main manuscript with the corresponding reference.

2) the reviewer requests a stability analysis. I think this would be a nice add on and increase the value of your paper, but also recognise that it could be considerable work. I leave it up to you as to whether you would like to include this or not.

We have included an additional analysis to determine if there is a significant difference in inter-annual variability at sites dominated by honeybees across all years, versus sites dominated by other species of insect pollinators. The t-test used is detailed in lines 171-174 of the methods section; the results provided in lines 231-233; and also referred to in the discussion in lines 272-273.

3) explain the issue with 'crop species' as identified by the reviewer

The reviewer was concerned that crop species was confounded with study ID and therefore it was best not used as a fixed effect in any of our models. As suggested by the reviewer, crop species has now been excluded from all the models and the text and tables referring to models previously containing crop species removed from the main manuscript as well as the supplementary material.

4) make the model statement as requested

The model statement has been made as requested in line 158

5) make amendments around the section on heterogeneity/stability as requested

The suggested amendments have been made and sentences speculating about the effect of landscape heterogeneity on stability (previously lines 268-270) have been deleted as requested by the reviewer.

Reviewer(s)' Comments to Author:

Referee: 1

Comments to the Author(s).

I find the revised version much improved and I am happy to see that most of my suggestions have been carefully addressed. I have only minor comments and a potential major suggestion that I missed from the first round of review (sorry!).

We are pleased to see the positive response from the reviewer and thank both reviewers of the previous manuscript version for their very constructive comments and feedback.

Starting from the latter, by reading L. 272-286 I was wondering if you could test the effect on stability of having the honeybee as the dominant species. You could classify the studies in two categories according to the identity of the dominant species (honeybee vs. wild bees). I imagine that for the same study you might have years where you have different dominant species but there should be a clever way to split studies to test the species identity effect. This could be a nice add-on.

We have now classified the study sites into two categories. One where honeybees are the dominant species across all years and the others where other insect pollinator species dominate. We ran a two-sample t-test to test for difference in inter-annual variation in pollinator abundance and found that the sites where honeybees were dominant has significantly higher stability. This add on analysis is detailed in lines 171-174 of the methods section; the results in lines 231-233 and also referred to in the discussion in lines 272-273.

Minor comments:

L.151 Here, I see that crop species is still included in the model. However, in the response letter the authors stated that they have excluded this variable from the modelling. I had suggested to omit this variable as crop species has too many levels and practically overlaps with study ID (your random). Please clarify.

We now excluded crop species from the models completely. Previously provided table S5 in the supplementary material has also now been removed to ensure that no models with crop species are featured anywhere in the main manuscript or the supplementary material.

L.164-165 Please state explicitly that you used ML and not REML to estimate your models in the multi-model inference analysis.

We have now explicitly stated that ML was used to estimate models and this information is provided in line 158.

L. 268-270 Please remove speculations about the effect of landscape heterogeneity on stability. This statement is not supported by your data.

Previously Lines 268-270 (now indicated as deleted lines 264-267) have been deleted to remove speculation about the effect of landscape heterogeneity on stability.